# Effect of Mobile-Based Counselling on Breastfeeding in Spain: A Randomized Controlled Trial Protocol (COMLACT Study)

**DOI:** 10.3390/healthcare11101434

**Published:** 2023-05-15

**Authors:** Desirée Mena-Tudela, Francisco Javier Soriano-Vidal, Rafael Vila-Candel, José Antonio Quesada, Laia Aguilar, Cristina Franco-Antonio

**Affiliations:** 1Department of Nursing, Feminist Institute University Institute for Feminist and Gender Studies, Universitat Jaume I, 12071 Castellon de la Plana, Spain; 2Department of Nursing, Universitat de València, 46007 Valencia, Spain; 3Department of Obstetrics and Gynecology, Xativa-Oninyent Health Department, 46800 Xativa, Spain; 4Department of Obstetrics and Gynecology, Hospital Universitario de la Ribera, 46600 Alzira, Spain; 5Department of Clinical Medicine, Universidad Miguel Hernández, 03202 Elche, Spain; 6Network for Research on Chronicity, Primary Care and Health Promotion (RICAPPS), 03550 Alicante, Spain; 7Midwifery at Lactapp Women Health, 08011 Barcelona, Spain; 8Department of Nursing, Universidad de Extremadura, 10003 Caceres, Spain

**Keywords:** exclusive breastfeeding, breastfeeding, e-Health, mobile application, midwives, nursing

## Abstract

Purpose: The primary aim of this study is to determine the influence of an intervention in women based on a free mobile application (LactApp^®^, Barcelona, Spain) in maintaining breastfeeding (BF) up to 6 months postpartum. The secondary aim is to assess the effect of health literacy (HL) on breastfeeding duration. Methods: A multicenter, randomized controlled clinical trial of parallel groups will be carried out. Women will be randomly assigned to each of the parallel groups. In the control group, usual clinical practice will be followed from the third trimester of pregnancy to promote BF. In the intervention group, and in addition to usual clinical practice, the women will use a free mobile application (LactApp^®^) from the third trimester to 6 months postpartum. The type of BF at birth, at 15 days and at 3 and 6 months postpartum and the causes of cessation of BF in both groups will be monitored. The hypothesis will be tested using inferential analysis, considering an alpha of 5%. The study protocol was approved by the Clinical Research Ethics Committee of Hospital de la Ribera (Alzira, Valencia, Spain) in February 2021. A per protocol analysis and an intention-to-treat analysis will be performed. Discussion: This study will identify the influence of a mobile application on improving BF rates. If the application proves effective, we will have a tool with free information available to any user at any time of day, which may be complemented by normal clinical practice and be integrated into our health care system. Trial registration: ClinicalTrials.gov ID: NCT05432700.

## 1. Introduction

The World Health Organization (WHO) recommends exclusive breastfeeding (EBF) for the first six months of an infant’s life, followed by complementary feeding until the infant is two or more years old [1]. The WHO firmly supports this type of feeding due to the clear benefits of breastfeeding (BF) for both the infant and the mother [2,3,4].

In Spain, the EBF rates at 6 months fall well short of the international recommendations [5]. Different studies have found the BF rates to be about 70% at maternal discharge, 68.4% at 6 weeks of life, and 24.7% at 6 months postpartum [6], though other more local studies report even lower rates [7]. A recent study on the prevalence of EBF up to 6 months of age carried out in our setting [8], involving a selected sample of women with an early start of BF in 87.7% of the cases, documented an EBF rate of 38.6% at 6 months postpartum, which is far below the worldwide target of 50% EBF at 6 months established by the WHO for the year 2025 [1].

A great variety of beneficial methods for promoting BF have been evaluated [9]. A number of health interventions have shown improvement of the BF rates, with proactive measures and the earliest intervention possible being the most successful strategies [10]. Both individualized and personalized support, as well as telephone mediated supportive measures, have demonstrated favorable outcomes [11,12], and the combination of various of these interventions enhances the effects and outcomes [13,14].

At present there are new interventions that still need to be tested for their long-term efficacy, such as the use of mobile applications [15,16]. In this context, mobile health services, mHealth, are defined as medical and public health practices compatible with mobile devices such as smartphones, patient monitoring devices, personal digital assistants and other wireless devices [17,18]. A large proportion of the European population consults online information for its health problems [19]. These users consider that the information they find is going to be useful for them [20]. On the other hand, healthcare systems increasingly make use of online services in support of users and healthcare professionals, including the utilization of mobile applications [21]. Nevertheless, it must be taken into account that the use of such tools is not proportional within the population, and that we must ensure that the information provided is not only based on the best available scientific evidence, but is also accessible to women at risk of social exclusion [22]. In line with different studies, the application of a protocolized intervention that is accessible and adapted to the needs of women effectively improves the rates and duration of BF [4,13,14,23,24,25,26].

On the other hand, while there is no single definition of health literacy (HL), the term refers to user knowledge and skills in adequate decision making in sociosanitary care [27]. Such skills include reading, writing, making calculations, communication capacity and—increasingly so—the use of information technologies. Deficient HL has been related to poor health outcomes, including the early suspension of BF [28,29,30], which is one of the most relevant public health problems and is of special importance in the maternal–childhood population.

In Spain, few randomized clinical trials have analyzed the benefits of interventions based on mobile devices in BF. The main objective of this study is to determine the influence of an intervention program in maintaining BF up to 6 months postpartum. The secondary aim is to assess the HL level of the participants and its association to the early (less than 6 months postpartum) suspension of BF.

## 2. Materials and Methods

### 2.1. Study Design

A multicenter, randomized controlled trial (RCT) of parallel groups has been designed. The clinical trial will be described according to the Consolidated Standards of Reporting Trials (CONSORT) [31] and using the Template for Intervention Description and Replication checklist (TIDieR) [32]. The study has been registered on ClinicalTrials.gov Identifier: NCT05432700.

### 2.2. Recruitment

The study will be conducted in three public hospitals in eastern Spain (Hospital Lluis Alcanyis, Hospital General de Castellón, and Hospital Universitario de la Ribera), and in one hospital in the western part of the country (Hospital San Pedro de Alcántara de Cáceres). The four hospitals serve a total population of 500,000 inhabitants, with an average of 5000 births a year. The women will be included during the third trimester of pregnancy in the different primary care midwife consulting rooms of each of the participating centers.

The study exclusion criteria are: females under 16 years of age; women with cognitive impairments, language barriers, or illiteracy (not able to read Spanish); non-availability of a mobile device with internet connection; newborn infants with congenital malformations; twin or multiple pregnancies; and admission to the Neonatal Intensive Care Unit (NICU), prenatal death or stillbirth, or postpartum complications requiring admission of the mother to the intensive care unit (ICU). Women failing to respond to the automated messages from the platform after three attempts will also be excluded, in the same way as those women in the control group who have used the LactApp^®^ application on their own initiative. Figure 1 shows the monitoring to be carried out during the study.

### 2.3. Randomization and Blinding

The women will be included during the third trimester of pregnancy in the different primary care midwife consulting rooms of each of the participating centers. After reading the information sheet and giving informed consent, the women will be registered on a webpage created for the purpose of the research and will be assigned to one of the groups (control or intervention) via a simple randomization process, until the estimated sample size has been reached. Randomization will be performed using Epidat v.4.2. [33], with groups of equal size.

Allocation concealment will be carried out, with blinding of the midwives in charge of recruiting the women, providing them with sealed, opaque and sequentially numbered envelopes to be delivered to each new participant wishing the participate in the study.

The research team will prepare the envelopes based on the random allocation sequence. Each envelope will contain a card with a QR code providing access to the study follow-up platform. Follow-up data collection will be conducted by another researcher blinded to the group status of participants. Only the data manager will have access to all of the integrated data (the data reported by the participating women through the surveys and the birth/postpartum data reported by the principal investigators).

### 2.4. Sample Size Determination

Based on previous studies [15], and assuming the BF suspension rate at 6 months to be 60% in the control group versus 40% in the intervention group, with a significance level of *p* < 0.05 and a statistical power of 85%, a balanced design would require 333 women in total. Compensating for an estimated loss rate of 20%, the estimated final total sample size would be 399 women.

### 2.5. Intervention

#### 2.5.1. Both Groups

The study sample will be compiled during the third trimester of pregnancy in the different primary care midwife consulting rooms of each of the participating centers. After receiving the envelope, the women will read the QR code with their mobile device and will be registered in the nursing platform created for the study, after accepting the privacy policy statement. Both the intervention group and the control group will undergo an initial survey to record sociodemographic data, previous experience with BF, and the probable date of delivery. In addition, two self-administered questionnaires will be completed—one on HL (HLS-EU-Q16) [34] and another on health-related quality of life (EQ-5D-5L) [35]—through the e-mail account registered on the platform.

Planning of reminders via e-mail will be performed to obtain information referred to follow-up, at all times avoiding the provision of any extra information on BF. During pregnancy, based on the pregnancy protocol and usual clinical practice, parent role education and nursing workshops will be planned for both groups, with the availability of different manuals to be given to each pregnant woman [36].

The platform will send serial e-mails to both study arms (intervention and control) for follow-up on the type of nursing after birth, at 15 days, at 6 weeks, and at 3 and 6 months. Questions will be asked regarding the type of nursing and BF-related problems, and the breastfeeding self-efficacy scale—short form (BSES-SF) will be applied [37]. In the event of the suspension of BF, the reason for suspension will be recorded, along with the total breastfeeding time. At 3 and 6 months postpartum, new measurements will be made with the health-related quality of life questionnaire (EQ-5D).

#### 2.5.2. Control Group (Usual Care)

Usual care includes individual counseling on the benefits of maintaining BF during the first 6 months and on the introduction of supplementary foods. The mother should be seen at least 6 times by the midwife and primary care pediatrician before the infant is 6 months old.

#### 2.5.3. Intervention Group (Mobile Application Group)

From the first recording during the first trimester, the women will be able to consult all the available information in the application. LactApp^®^ is a free access mobile application developed through Apple Store and Google Play, and can be consulted in English and Spanish [38,39]. The central function of LactApp^®^ is its automated nursing consultation structure. This tool also uses artificial intelligence and is available 24 h a day with a connection to the Internet, providing convenient and personalized support. LactApp^®^ includes a self-management form based on over 50 decision trees with questions and answers developed by nursing experts, and is supported by scientific evidence and current official health recommendations. The questionnaire generates over 2300 personalized replies that can be reached through over 76,100 potential routes that vary fitting to the profile of the users and the options selected by the latter. In addition, the application reminds the user about the following issues, depending on the entered date of birth:The first four weeks: positioning of the infant for breastfeeding, frequency of milk intake, number and consistency of stools, general care of the breast, and weight gain of the newborn. For women providing food supplements, the tool offers counseling and support for returning to breastfeeding.From months two to three: recommendations on milk extraction to create a reserve in case of having to return to work or leave home, with instructions on handling and storage of the collected milk.From months four to six: how to use the milk reserve (if any) and techniques for administering the stored milk, placing emphasis on the importance of supporting BF and the advisability of not using other types of feeding.

If the intervention is effective and differences are found between the two groups, an analysis will be conducted within the intervention group to analyze the profile of women who are less likely to abandon BF at 6 months based on baseline variables, by adjusting a multiple logistic regression model, estimating the corresponding ORs and 95%CIs.

### 2.6. Data Collection and Follow-Up

The collected data will be entered into an electronic form, guaranteeing confidentiality and anonymity, and ensuring compliance with the applicable regulations. Likewise, losses and dropouts during the trial will be detailed, along with the corresponding causes. In addition, e-mails will be sent to the mothers, inviting them to follow-up on breastfeeding using the self-administered questionnaire found on the created web platform.

The principal investigators will have access to the body of filtered data on the website created for the project, protected by a password, to be able to retrospectively enter the birth data and the number of postpartum visits to the different health professionals. In order to guarantee confidentiality, the data supplied to the project team members will not contain information capable of identifying the participants.

### 2.7. Baseline Variables

The following variables will be collected at baseline:

- Sociodemographic variables: maternal age at the probable date of delivery, country of origin (Spain/foreign), level of education (primary school or lower/secondary school/university), employment status (self-employment/professional/managerial employment/employee/unemployed/student/not looking for a job), civil status (single/married/separated-divorced), partner (yes/no), number of live offspring, previous experience with breastfeeding (yes/no), participation in maternal education groups (yes/no), reception of previous information on BF (yes/no), and participation in BF support groups (yes/no).

- Variables related to health literacy: as a screening tool, use will be made of the HLS-EU-Q16, which evaluates the HL of the population based on 16 items scored by means of a Likert scale from “very easy” to “very difficult”. This is a unifactorial scale with good internal consistency as measured by McDonald’s omega, with a value of 0.982 in the Spanish population [34].

- Variables referred to health-related quality of life: use will be made of the EQ-5D [40] in the baseline measurements and at 3 and 6 months. This scale is very sensitive to quality of life measures and has been shown to be even more sensitive in women [41]. It comprises 5 items related to mobility, personal care, daily activities, pain/discomfort, and anxiety/depression. It uses a visual analogue scale (VAS) from 0 to 100 for measuring health condition at the time of application of the questionnaire [42].

- Variables related to self-efficacy of BF: the measurement of self-efficacy will be made with the BSES-SF scale in its Spanish version [37]. This instrument consists of 14 items scored by means of a Likert scale from 1 (not at all) to 5 (always). The BSES-SF is a unidimensional scale with a Cronbach alpha of 0.79 in its Spanish version.

- Obstetric-neonatal variables. The following information will be collected retrospectively from the electronic case history: gestational age (days) at the time of birth, parity (none/one or more), type of previous deliveries (none/eutocic/other), onset of labor (spontaneous/induced/stimulated), rupture (spontaneous/artificial), group B streptococcus (positive/negative), intrapartum antibiotic use (yes/no), intrapartum analgesia (inhalatory/local/epidural/none), Kristeller maneuver (yes/no), completion of delivery (eutocic vaginal/eutocic instrumental (vacuum, spatulas, forceps), intrapartum cesarean section), episiotomy (yes/no), perineal condition following birth (intact/grade 1/grade 2/grade 3/grade 4), newborn gender (female/male), newborn weight (grams), Apgar score, umbilical arterial pH at birth, early skin-to-skin contact (within 30 min and lasting for at least 2 continuous hours) (yes/no/with father), early start of breastfeeding (within 2 h/after more than 2 h), drinking allowed during delivery (yes/no), accompaniment of maternal choice allowed (yes/no), mobilization allowed during delivery (yes/no), and positioning in moment of birth (vertical/lying down—lithotomy position/lateral decubitus).

- Response variable: type of nursing (BF/SF/MF) at 6 months postpartum, to assess newborn and infant feeding practices, with the following options: (a) BF, including extracted or donor milk. The infant only receives drops or syrups (vitamins, mineral, medicines); (b) SF (supplementary feeding), where infant feeding is limited to artificial formulas and solid; and (c) MF (mixed feeding), where infant feeding combines BF and SF. All women intend to offer BF after birth. The response variable “Suspension of BF at 6 months” (yes/no) will be considered, where “yes” means the infant is receiving SF and “no” means the infant continues with BF or MF at 6 months.

- Variables related to suspension of BF: type of suspension (total or partial), cause of early suspension (before 6 months postpartum), and total duration of BF in full days from time of birth.

- Follow-up variables: participation in support groups, and number of midwife/pediatrician/pediatric nurse visits during the first 6 months. Even if women suspend BF, the number of visits to these health professionals will continue to be counted for up to 6 months.

- Variables related to BF education received for 6 months: information/training in BF (none/previous information received from relatives, friends or health professionals (midwife, pediatric nurse, obstetrician, pediatrician)), consultation of texts, participation in birth preparation groups, nursing groups or postpartum groups, and the use of digital tools (yes/no/specify which).

- LactApp^®^ use will be assessed from women’s reports on the acceptability and convenience of the mobile application while during use. Data will be collected on the number of connections made, time of use, and topics consulted. All of these data will be provided by the company in an anonymized form. No private information resulting from the mobile application will be collected.

### 2.8. Data Analysis

An analysis will be performed of the baseline characteristics between the group of possible losses to follow-up and the group of women that complete follow-up, based on two-input tables and the comparison of means, using the Fisher exact test or Mann–Whitney U-test, as applicable.

Testing of the homogeneity of groups will be completed regarding the baseline variables and birth-related parameters and the suspension of BF at 6 months, using two-input tables and the chi-square test (χ^2^) or Fisher exact test, as applicable, while the comparison of means will be performed using the Mann–Whitney U-test.

A per-protocol (PP) analysis and intention-to-treat (ITT) analysis will be carried out. In the PP analysis, continuous variables will be reported as the mean and standard deviation (SD), or as the median and interquartile range (IQR) in the case of a non-normal data distribution, while categorical variables will be reported as frequencies and percentages. Data normality will be assessed using the Kolmogorov–Smirnov test.

In the ITT analysis, sensitivity referring to losses during follow-up will be assessed, with the response variable of the intervention group being assigned the worst result possible (suspension of BF at 6 months) and the control group being assigned the best result possible (no suspension of BF). Evaluation of the differences in response variable between the two groups will be performed, with all randomized subjects, applying the chi-square test.

#### 2.8.1. Primary Outcome

The efficacy of the intervention in relation to the suspension of BF at 6 months will be assessed from two-input tables with application of the chi-square test (χ^2^). Likewise, multivariate logistic regression analysis will be used, with calculation of the odds ratio (OR) corresponding to the suspension of BF at 6 months, and the 95% confidence interval (95%CI).

#### 2.8.2. Secondary Outcome

The level of HL (adequate vs. inadequate) with the HLS-EU-Q16 screening tool and the characteristics of the women who gave up BF at 6 months (yes/no) will be assessed via two-input tables with application of the chi-square test (χ^2^) for qualitative variables. Multivariate logistic models will be used in order to analyze the magnitude of the association with BF at 6 months. The adjusted odds ratio (OR) and 95% confidence interval (95%CI) will be calculated, and *p*-values will be presented.

Data analysis will be performed using SPSS v.26.0 for Windows (IBM Corp. 2018, Armonk, NY, USA) and R (R project 2019, version 4.0.2). The level of statistical significance defined was *p* < 0.005.

### 2.9. Ethical Considerations

The protocol is registered in ClinicalTrials.gov ID: NCT05432700. The study will abide by Organic Act 3/2018, of 5 December, referring to personal data protection and the guarantee of digital rights, and by Act 41/2002, of 14 November, regulating patient autonomy and the rights and obligations in relation to information and scientific documentation. Likewise, the ethical principles of the Declaration of Helsinki will be followed. Informed consent will be obtained from all the participating women. LactApp^®^ complies with the policies referring to privacy and cookies, guaranteeing abidance with the measures required by Regulation (EU) 2016/679 of the European Parliament and Council, of 27 April 2016, regarding the protection of physical persons with regard to to personal data and the free circulation of such data, Organic Act 15/1999, of 13 December with regard to data protection (LOPD), and Spanish Royal Decree 1720/2007, of 21 December, with due declaration before the Spanish Data Protection Agency. Likewise, the study will abide by Act 34/2002, of 11 July, on electronic commerce and information regarding the use of cookies (LSSICE), expressly requiring consent from the users registered to the application.

### 2.10. Validity and Reliability

The entire dataset will be confirmed after collection, and any subjects found will be directly amended by the researcher. The identification of obvious errors and outliers in the data will be shown by descriptive analysis. When necessary, data will be double-checked against the original databases.

Factors such as age, education level, employment situation, and previous experience with BF may influence the results of the study. Due to randomization, decompensated key variables between groups are not expected. However, if poorly distributed characteristics are found, these variables will be included as independent variables along with the exposure variables in the multivariate analysis models.

## 3. Discussion

The purpose of this trial is to explore the impact of an intervention program based on the use of a mobile application in maintaining breastfeeding up to 6 months postpartum. The null hypothesis is that the LactApp^®^ mobile application does not improve the maintenance of BF at 6 months postpartum versus usual clinical practice.

Breastfeeding is not only a physiological process but also entails cultural learning, causing mothers to actively seek information [19]. Nowadays, online communication options have been incorporated into this active search for information [24], and the adoption of such new technologies has been associated with improved nursing rates [43,44]. Although not under professional influence, the use of information and communication technologies in daily life must be acknowledged, and such tools should be used as a means to disseminate quality information based on evidence [45]. In 2018, 67.3% of the Spanish population used the Internet to obtain information on health, and in this regard, mobile applications constitute a growing contribution in the field of information and communication technology [45]. Health professionals must make efforts to consolidate their position as a reference in these new settings as well. If they prove effective and acceptable to the population for which they are intended, instruments of this kind could be introduced as intervention tools for promoting and maintaining BF.

An important percentage of users increasingly take an active role in issues related to health and illness and their relation to the sociosanitary system [46], leaving behind the passive or mere spectator roles of the past [47]. In this context, it is a fact that pregnant women comment to their midwife what they have read on the Internet, and wish to discuss such information [45]. At present, three out of every four users consult the Internet with regard to health issues. This process of empowerment defines the “expert patient”, who seeks information both before and after visiting a healthcare professional [48]. Thus, it must be accepted that in contemporary maternity care, women routinely and massively use online sources to obtain information about their pregnancy, and consequently the maternal–infant health services should adopt strategies that take this phenomenon into account [49]. Such consideration also applies to breastfeeding, where the tendencies are changing, and women now seek online information about infant feeding just as often as they tend to consult nearby family and friends [50], receiving both online support and information through social networks, video calls, or mobile applications [51]. A recent systematic review and meta-analysis [15] of 15 randomized controlled trials (RCTs) of mHealth interventions for BF found that these interventions significantly increased EBF rates at 1, 2, 3, and 6 months postpartum and improved BF self-efficacy, but not attitudes. Updated applications are needed in the latest recommendations on nursing [49,52], with external evaluation guarantees.

Thus, the traditional care model must integrate electronic devices and information and communication technologies to ensure improved performance, since reliance upon healthcare professionals could decrease, giving way to the availability of interactive information through the different technological platforms [15,53]. Health-related applications, mobile phones, blogs, and specific websites are typically used by women of childbearing age and may help improve the experience of pregnancy and motherhood. It is essential for both professionals and health supervisors to be aware of these new developments, which can influence the health of women, and to anticipate the imminent change in care model [54,55].

## 4. Limitations

The present study has its limitations. The level of adherence or decisions of the pregnant women when the issue of breastfeeding is raised during the monitoring of pregnancy may depend on a number of factors such as the type of information given to the women and the way in which the health professionals communicate with them. In effect, we consider that favorable health outcomes to be largely dependent upon the complex interaction between the health professional and the patient or user. However, the aim of our study is to explore the patient-related factors that facilitate or complicate BF, such as poor health literacy, with reference to the body of literature cited in the protocol. Since there is currently no standardized and protocolized mechanism for determining the skills of pregnant women in making effective use of the information provided, we consider it prudent to assume that the information supplied will be more or less the same for each woman (LactApp^®^ + professional), and that women with limited HL will be more prone to making suboptimal decisions as a result. In view of these shortcomings, we feel that it will be difficult to find compensation mechanisms for these women. Logically, this does not obviate the individual efforts which the professionals may make to solve problems such as poor understanding or language issues. Nonetheless, we believe that the added complexity of trying to systematize and define such efforts falls beyond the scope of our study. Furthermore, a contamination effect could occur among the women participating in the study, relating to the exchange of information regarding the use of the mobile application. This could be detected through the periodic electronic nursing follow-up surveys, evaluating whether the women in the control group make use of online tools. Such use would be reason for exclusion from this study. Another limitation may be the number of losses over follow-up. The self-completed electronic survey via the Internet has many advantages such as speed of data collection, lower cost, and can provide an attractive design for the respondent [41]. The downside is that there may be a lower response rate, which may affect the results obtained. In order to control for such selection bias and break of randomization, an analysis will be performed of the characteristics of the group of dropouts, along with an intention-to-treat analysis.

Finally, it should be noted that there are limitations related to the characteristics of the sample. The results cannot be extrapolated to specific populations such as multiple pregnancies, newborns admitted to the neonatal intensive care unit, or women who experience complications in the postpartum period because of the difficulty of initiating breastfeeding. On the other hand, our sociocultural context and common access to mobile devices in our environment will limit the extrapolation of results to other populations with marked differences.

## 5. Conclusions

After conducting this trial, it will be possible to evaluate the breastfeeding rate 6 months postpartum among women who have used the mobile application.

If the search for information on health-related topics increases their consultations, we should at least consider that users need to expand or contrast the information received. It is important that we become involved in the creation of digital content so that the information received by users is truthful, complementary and verified. If the application proves to be effective, we will have a tool with information free of charge, available to any user, at any time of the day, which can be complementary to normal clinical practice and can be integrated into our healthcare system. For all of these reasons, the project demonstrates its translational value, since the basis of the intervention is a free and openly accessible mobile application that could also be used in other areas.

## Figures and Tables

**Figure 1 healthcare-11-01434-f001:**
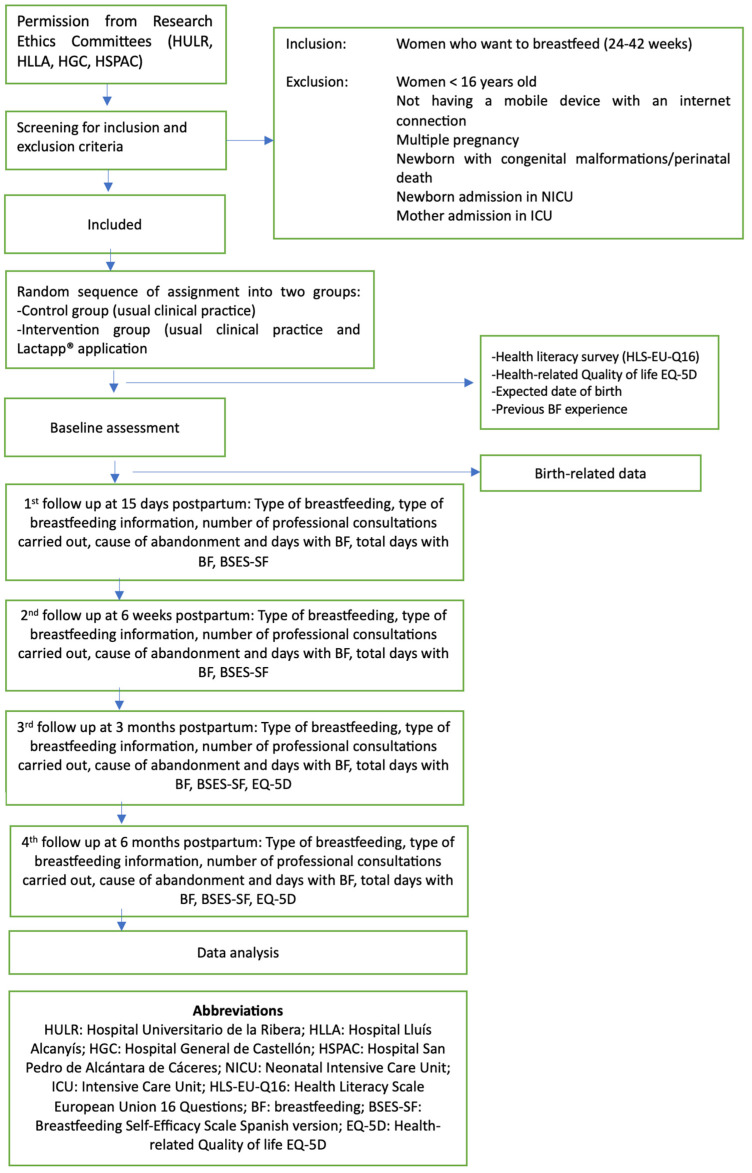
Study design and interventions flowchart.

## Data Availability

Data are available upon reasonable request. All necessary data are supplied and available in the manuscript; however, the corresponding author will provide the dataset upon request.

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
