# Peer review of "Effect of Mobile-Based Counselling on Breastfeeding in Spain: A Randomized Controlled Trial Protocol (COMLACT Study)"

_healthcare, 2023, doi:10.3390/healthcare11101434_

Round 1
Reviewer 1 Report
Review of Healthcare-2305958 (Effect of the use of a mobile application ….)
I am a bit confused as to the name of the journal, as the copy I got said ‘IJERPH,’ but the request said ‘Healthcare.’
Please add “Spain” to the title. Studies of this sort can be heavily dependent on social context.
Throughout the protocol, it seems that “BF” and “EBF” are used almost interchangeable. Please be very precise about this. As this protocol stands, it seems that the only outcome is EBF at 6 months (or is it any BF at 6 months?). My understanding is that in general, the longer (up to 6 months) the better, so that stopping EBF at 2 months is worse than stopping at 4. This might imply that a survival analysis would be more appropriate.
Specific comments
line comment
1 62 “important”—Do you mean “successful,” “commonly used”?
2 111 “global”—I think “total” is better.
3 112 This is the first of many uses of “selected.” It is not clear to me in what way, other than being eligible for the study and consenting to take part, women will be selected. Are they going to ask everyone who might qualify, then accept all that do, until they reach the desired sample size? Or is something else going on? If the extent of “selection” is as I describe it above, please find a different word (“enrolment” perhaps, but then again, maybe you don’t want to formally enroll until after delivery).
4 119 “failing to respond”—when? Ever? The first one? I would prefer to see this trial emulate real life, if only because that will decrease the likelihood of informative exclusions (i.e., that those excluded will be those most likely to be unsuccessful in the trial). As for excluding “control” women who access the app, I’m afraid that this will lead to a lot of exclusions. To some extent it is interesting to know how much women will find out on their own (though I understand that some things may come up in conversation in the waiting room). Please think carefully about what the comparison of interest is and about how various types of exclusions after starting the trial will affect the inference. To some extent, you want to know whether a program of promoting the mobile app is better than a poster in the clinic or occasional remarks from the midwives. You might add a question at the 6 month visit asking about whether they used the app, and if so, how they found out about it.
5 Fig. 1 Please add gestational age or postpartum time, where appropriate. This will make it easier for readers to understand.
6 127 “obtainment”—“giving” would be better (it is the women who are reading the information sheet and giving informed consent. Besides, “obtainment” is not an English word.
7 130 “established”—should probably be “reached.”
8 155 Has the abbreviation “HL” been defined before?
9 186 Please change “user” to “users.”
10 208-21 Why will the principal investigators need to see the filtered data?
11 214-218 Variables I did not see: educational attainment, work outside the home, urban/rural. Please think about the types of variables known to affect BF and EBF. Also consider testing for effect modification by some of the sociodemographic variables (e.g., education, urban/rural, Spanish vs. foreign birth come to mind).
12 249-256 Does “AF” include solids such as rice cereal or mashed banana? I don’t think these would normally be considered “artificial.” Maybe you need a better word, such as supplementary. Or maybe people in Spain don’t give mushy solids to babies under 6 months.
13 262-265 Please note that for counts of events (such as how many visits people had) happening during the trial, those who stop EBF early (and are therefore done with the trial) are likely to have fewer encounters, simply by being in the trial for a shorter period. A better measure is something like the fraction of encounters actually had divided by the number the woman should have had for the time
Author Response
Response to Reviewer 1 Comments
Thank you for your very positive and constructive feedback on our manuscript.
We have considered all your comments and suggestions (and also the comments made by the other reviewer) attempting to improve/refine the original manuscript.
Below you will find a point-by-point response to your comments (in red).
Point 1: I am a bit confused as to the name of the journal, as the copy I got said ‘IJERPH,’ but the request said ‘Healthcare.’
Response 1: Thank you for your comment. The date of submission to the journal IJERPH was on 03/07/23. Due to the notification, we received from the Assistant Editor on 03/22/23 about the non-indexing of the IJERPH journal in the Web of Science database, we decided to transfer it to Healthcare as proposed. The Section Managing Editor was in charge of the transfer process. Indeed, we have verified that the template in .doc format is still that of IJERPH. In this new revised version, we have adapted it to the correct format.
Point 2: Please add “Spain” to the title. Studies of this sort can be heavily dependent on social context.
Response 2: Thank for your highlight. A new title has been rewritten.
Point 3: Throughout the protocol, it seems that “BF” and “EBF” are used almost interchangeable. Please be very precise about this. As this protocol stands, it seems that the only outcome is EBF at 6 months (or is it any BF at 6 months?). My understanding is that in general, the longer (up to 6 months) the better, so that stopping EBF at 2 months is worse than stopping at 4. This might imply that a survival analysis would be more appropriate.
Response 3: Thank for your comment. We agree with the comment. The aim of the study is to evaluate breastfeeding (BF) rather than exclusive breastfeeding (EBF) due to the low prevalence of EBF at 6 months in Spain. EBF has been changed to BF in the summary section (line 32), sample size determination (line 132), intervention Group (line 187), baseline variables (lines 239-246), and data analysis (has been rewritten).
Regarding the comment on survival analysis. A survival analysis models the time to the occurrence of an event, which in this case may be the suspension of BF. Both Cox models and Kaplan-Meier curves require continuously measured times. In this trial, the times are discrete, measured at fixed visits from the time of delivery, with the final fixed time of 6 months being the target of the study. It is not possible to apply survival analysis, as we do not know the exact time of suspension of breastfeeding.
Point 4: Line 62 “important”—Do you mean “successful,” “commonly used”?
Response 4: Thank for your highlight. Amended in line 52.
Point 5: Line 111 “global”—I think “total” is better.
Response 5: Thank you for your comment. Amended in line 100.
Point 6: Line 112 This is the first of many uses of “selected.” It is not clear to me in what way, other than being eligible for the study and consenting to take part, women will be selected. Are they going to ask everyone who might qualify, then accept all that do, until they reach the desired sample size? Or is something else going on? If the extent of “selection” is as I describe it above, please find a different word (“enrolment” perhaps, but then again, maybe you don’t want to formally enroll until after delivery).
Response 6: Thank for your highlight. The term “selected” has been changed by “included” in lines 101 and 115.
Point 7: Line 119 “failing to respond”—when? Ever? The first one? I would prefer to see this trial emulate real life, if only because that will decrease the likelihood of informative exclusions (i.e., that those excluded will be those most likely to be unsuccessful in the trial). As for excluding “control” women who access the app, I’m afraid that this will lead to a lot of exclusions. To some extent it is interesting to know how much women will find out on their own (though I understand that some things may come up in conversation in the waiting room). Please think carefully about what the comparison of interest is and about how various types of exclusions after starting the trial will affect the inference. To some extent, you want to know whether a program of promoting the mobile app is better than a poster in the clinic or occasional remarks from the midwives. You might add a question at the 6 month visit asking about whether they used the app, and if so, how they found out about it.
Response 7: Thank you again for your input. In line 108 we have specified at three attempts. This same reflection has already been discussed among the research team. We decided to exclude women who did not respond to any of the surveys because non-response could be interpreted in two different ways: a) she has suspended BF, or b) she is still with BF but decides not to answer.
We know that the loss of sample may be important but keeping these women in the study could negatively affect the results. On the other hand, we also decided to eliminate women in the control group who use the app because we would not know whether the effect of BF maintenance is produced by the midwife's advice or not, or by a combination of both. Regarding the possibility of including a question at 6 months about whether she has used the app, this will be done, as indicated in the baseline variables section, line 261. We include in this new version specify which (line 258), and for 6 months (line 254).
Point 8: Fig. 1 Please add gestational age or postpartum time, where appropriate. This will make it easier for readers to understand.
Response 8: Thank you for you comment. Amended in page 4.
Point 9: Line 127 “obtainment”—“giving” would be better (it is the women who are reading the information sheet and giving informed consent. Besides, “obtainment” is not an English word.
Response 9: Thank you for your comment. Amended in line 117.
Point 10: Line 130 “established”—should probably be “reached.”
Response 10: Thank you for your improvement. Amended in line 120.
Point 11: Line 155 Has the abbreviation “HL” been defined before?
Response 11: Thank you for your comment. The abbreviation is in line 77 when the term health literacy first appears.
Point 12: Line 186 Please change “user” to “users.”
Response 12: Thank you for your comment. Amended in line 176.
Point 13: Line 208-21 Why will the principal investigators need to see the filtered data?
Response 13: Thank you for your comment. The principal investigators need review the birth data/visits of health professionals and introduce on the website of RCT. A new paragraph has been written in lines 196-197.
Point 14: Line 214-218 Variables I did not see: educational attainment, work outside the home, urban/rural. Please think about the types of variables known to affect BF and EBF. Also consider testing for effect modification by some of the sociodemographic variables (e.g., education, urban/rural, Spanish vs. foreign birth come to mind).
Response 14: Thank for your highlight. We fully agree with the comment. By mistake we have omitted the other socio-demographic variables that have been included in the study. A new paragraph has been presented in lines 203-206.
Point 15: Line 249-256 Does “AF” include solids such as rice cereal or mashed banana? I don’t think these would normally be considered “artificial.” Maybe you need a better word, such as supplementary. Or maybe people in Spain don’t give mushy solids to babies under 6 months.
Response 15: Thank you for your comment. In order to improve the understanding of the differences, we have included the term supplementary feeding instead of artificial feeding, in lines 239-246.
Point 16: Line 262-265 Please note that for counts of events (such as how many visits people had) happening during the trial, those who stop EBF early (and are therefore done with the trial) are likely to have fewer encounters, simply by being in the trial for a shorter period. A better measure is something like the fraction of encounters actually had divided by the number the woman should have had for the time
Response 16: Thank you for your comment. Women who suspension BF, do not drop out of the study, and will be followed to the end, therefore, do have their visits to medical services correctly measured.
In order to improve the understanding of this paragraph, we have included new text on the lines 251-253.

Reviewer 2 Report
1.Title can be modified as ' Effect of mobile based counselling on exclusive breast feeding- A randomized controlled trial'
2.Is abstract required for a protocol?
3.Introduction can be condensed. Aim is broad goal of the study while objective statement should be specific, observable and measurable. Suitable changes to be made in the introduction.
4.It should be 'Material and methods' and not 'Materials'
5.Consort flow chart can be improved as per guidelines. Manyetails can be deleted
6.How do you blind this type of study?
7.Word 'complimentary feeding' may replace 'supplementary'
8.Ethical consideration, variables and limitations can be made more specific
9. Manuscript can be condensed significantly
Author Response
Response to Reviewer 2 Comments
Thank you for your very positive and constructive feedback on our manuscript.
We have considered all your comments and suggestions (and also the comments made by the other reviewer) attempting to improve/refine the original manuscript.
Below you will find a point-by-point response to your comments (in red).
Point 1: Title can be modified as ' Effect of mobile based counselling on exclusive breast feeding- A randomized controlled trial'
Response 1: Thank you for your comment. We agree with the proposed title. In addition, we have included Spain at the request of another reviewer.
Point 2: Is abstract required for a protocol?
Response 2: Thank you for your comment. In other protocols published by the journal they have been required.
Point 3: Introduction can be condensed. Aim is broad goal of the study while objective statement should be specific, observable and measurable. Suitable changes to be made in the introduction.
Response 3: Thank you for your comment. The introduction has been structured in four distinct parts: 1) Benefits of breastfeeding; 2) Prevalence of breastfeeding in Spain; 3) Methods for the promotion of BF (eHealth); 4) Health literacy and breastfeeding.
We consider that we can reduce part 1(new paragraph lines 38-41), we believe that the rest provides a coherent discourse that justifies the RCT.
We believe that the study design, the methodology employed, and the planned statistical analysis will be able to observe and measure the proposed objectives.
Point 4: It should be 'Material and methods' and not 'Materials'
Response 4: Thank you for your comment but in https://www.mdpi.com/journal/healthcare/instructions, it is stated Materials and Methods.
Point 5: Consort flow chart can be improved as per guidelines. Manyetails can be deleted
Response 5: Thank you for your comment. Amended in page 4. We plan to use the Consort flow chart when we have the results. The purpose of the chart is to visually capture the women's entry into the study and the timing of the surveys and follow-up. The arrow error has been amended.
Point 6: How do you blind this type of study?
Response 6: Thank you for your input. The blinding of the study is explained in the section on randomization and blinding (lines 122-124). Blinding will be performed on the participating midwives, as described, so that they cannot influence the groups. The principal investigators will also not know which group each woman belongs to because this record is hidden in the designed web platform. Only the Data Manager will have access to all the integrated data (the data reported by the participating women through the surveys and the birth/postpartum data reported by the principal investigators). In order to improve this question, we have to introduced this explanation in a new paragraph in lines 128-130.
Point 7: Word 'complimentary feeding' may replace 'supplementary'
Response 7: Thank you for your comment. We agree with your comment. From the age of 4 months, the newborn can take solid or crushed food, and not only artificial milk. Amended in lines 239-246.
Point 8: Ethical consideration, variables and limitations can be made more specific
Response 8: We regret that we do not understand this comment. We would like you to specify in what direction you should be more specific. Due to the requirements of the journal, there is another complementary section after the conclusions, where the Institutional Review Board statement and Informed consent statement must be specified (lines 411-424).
Point 9: Manuscript can be condensed significantly
Response 9: We regret to understand the comment, where it refers, in addition to the indicated in the introduction.

Reviewer 3 Report
This manuscript is a protocol of an RCT investigating the beneficial effects of the use of mobile application as an adjuvant therapy to maintain breastfeeding. As eHealth becomes important, this topic is public health-related.
However, the following limitations are pointed out.
1) Critical: The total sample size in this manuscript is described as 214. However, their protocol lists 399 participants as their estimated enrollment. These discrepancies cause fatal flaws in clinical trials.
2) Figure 1 could be improved further. Please check the arrow errors in the current figure. Abbreviations are missing. The authors could improve their figure 1 by utilizing the CONSORT flow diagram.
3) Page 5, ‘2.5. Intervention’ includes not only contents of intervention, but also those of assessment.
4) The authors should describe the outcome of this clinical trial by dividing it into primary and secondary.
5) Please describe in more detail how to evaluate usability of the mobile application.
Author Response
Response to Reviewer 3 Comments
Thank you for your very positive and constructive feedback on our manuscript.
We have considered all your comments and suggestions (and also the comments made by the other reviewer) attempting to improve/refine the original manuscript.
Below you will find a point-by-point response to your comments (in red).
This manuscript is a protocol of an RCT investigating the beneficial effects of the use of mobile application as an adjuvant therapy to maintain breastfeeding. As eHealth becomes important, this topic is public health-related.
However, the following limitations are pointed out.
Point 1: Critical: The total sample size in this manuscript is described as 214. However, their protocol lists 399 participants as their estimated enrollment. These discrepancies cause fatal flaws in clinical trials.
Response 1: Thank you for your comment. This is indeed an error in the transcription of the sample size, 399 being the original and correct estimated total sample (lines 132-136).
Point 2: Figure 1 could be improved further. Please check the arrow errors in the current figure. Abbreviations are missing. The authors could improve their figure 1 by utilizing the CONSORT flow diagram.
Response 2: Thank you for your comment. We plan to use the Consort flow chart when we have the results. The purpose of the chart is to visually capture the women's entry into the study and the timing of the surveys and follow-up. The arrow error has been amended. Abbreviations are at the bottom of the figure (page 4).
Point 3: Page 5, ‘2.5. Intervention’ includes not only contents of intervention, but also those of assessment.
Response 3: Thank you for your comment. The assessment of intervention is included in analysis section (lines 283-295).
Point 4: The authors should describe the outcome of this clinical trial by dividing it into primary and secondary.
Response 4: Thank you for your comment. We have rewritten the data analysis section and we have grouped the assessment and the outcomes (primary and secondary) as proposed (line 283 and line 289).
Point 5: Please describe in more detail how to evaluate usability of the mobile application.
Response 5: Thank you for your comment. The term usability has been changed by use because usability implies evaluating the quality of the app, and in our case, because we are not the owners of the app we will only evaluate the number of connections, time of use and topics consulted. In order to improve this question, we have introduced a new paragraph in lines 259-263.

Round 2
Reviewer 1 Report
Review of Healthcare 2305958.R1
The protocol is much improved. I still have a few questions that need to be clarified.
line comment
1 100-103 To some extent, this comment deals with efficacy (real world) vs. effectiveness (whether it works in ideal circumstances). Given the type of intervention, I would like to see this trial deal with efficacy. Therefore, I hope that the ‘exclusions’ discussed in these lines will be from the time they happen, and that data until then will be used. As I read this sentence, failing to respond to 3 prompts refers to “at the beginning.” Obviously these women will have no outcome data, but I hope that the study plans to use their baseline data to examine risk factors for nonuse of the app (if the number of never responders is large enough). Control women’s data should certainly be included until they access the app.
2 160-161 Women not literate in either Spanish or English should not be eligible for the trial.3 243 “suspension” should be ‘stop” or “suspend.”
3 251-255 If the data from Lact-app are anonymous, how will the study know that control women accessed the app? Will they give the app a list of women in the study? Does that have privacy implications?
4 270-273 This is the most extreme way to deal with loss to follow-up.
5 282-287 Since this is a randomized trial, in theory, there is no need for looking at covariates. Nonetheless, this will be interesting and should be retained.
6 In addition, I suggest looking for effect modification of the trial arm effect by the baseline variables.
Author Response
Response to Reviewer 1 Comments Round 2
Thank you for your very positive and constructive feedback on our manuscript.
We have considered all your comments and suggestions.
Below you will find a point-by-point response to your comments (in red).
Point 1: 100-103 To some extent, this comment deals with efficacy (real world) vs. effectiveness (whether it works in ideal circumstances). Given the type of intervention, I would like to see this trial deal with efficacy. Therefore, I hope that the ‘exclusions’ discussed in these lines will be from the time they happen, and that data until then will be used. As I read this sentence, failing to respond to 3 prompts refers to “at the beginning.” Obviously, these women will have no outcome data, but I hope that the study plans to use their baseline data to examine risk factors for nonuse of the app (if the number of never responders is large enough). Control women’s data should certainly be included until they access the app.
Response 1: Thank you for your comment.
Point 2: 160-161 Women not literate in either Spanish or English should not be eligible for the trial.
Response 2: Thank you for your highlight. It is true, although we implicitly think about it, it is not clear in the inclusion/exclusion criteria. Now, has been included as an exclusion criterion, in lines 96-97.
Point 3: 243 “suspension” should be ‘stop” or “suspend.”
Response 3: Thank you for your correction. Amended in line 247.
Point 4: 251-255 If the data from Lact-app are anonymous, how will the study know that control women accessed the app? Will they give the app a list of women in the study? Does that have privacy implications?
Response 4: Thank you for your comment. Among the variables we will collect (digital tools in line 253), we will ask all women if during the 6 months they have received information on breastfeeding and specifically if they have used digital tools. The women themselves will be the ones to answer this question through the platform. The researchers will not have access to any of the women's private data, except for the Data Manager. The entire database will be given to the researchers for analysis in an anonymised form, so privacy will be respected.
Point 5: 270-273 This is the most extreme way to deal with loss to follow-up.
Response 5: Indeed, this ITT approach is the most unfavourable, ensuring that if there were significant differences between groups in the per-protocol analysis, and they were maintained in the ITT analysis, we would be confident of the efficacy of the intervention.
Point 6: 282-287 Since this is a randomized trial, in theory, there is no need for looking at covariates. Nonetheless, this will be interesting and should be retained.
Response 6: Thank you for your input.
Point 7: In addition, I suggest looking for effect modification of the trial arm effect by the baseline variables.
Response 7: Thank you for your highlight. A new paragraph has been added in lines 181-184 as follows:
If the intervention is effective and differences are found between the two groups, an analysis will be conducted within the intervention group to analyse the profile of women who are less likely to abandon BF at 6 months on baseline variables, by adjusting a multiple logistic regression model, estimating the corresponding ORs and 95%CIs.

Reviewer 2 Report
1Authors have carried out some changes but can be improved further\.
2.Letter 'a' can be removed from the title before mobile based
3.Introduction can be condensed. Some details can go to discussion section (3rd para- lines 56-77)
4.Limitation: can be made more specific rather describing
Author Response
Response to Reviewer 2 Comments Round 2
Thank you for your very positive and constructive feedback on our manuscript.
We have considered all your comments and suggestions (and also the comments made by the other reviewer) attempting to improve/refine the original manuscript.
Below you will find a point-by-point response to your comments (in red).
Point 1: 1Authors have carried out some changes but can be improved further.
Response 1: Thank you for your comment.
Point 2: Letter 'a' can be removed from the title before mobile based
Response 2: Thank you for your comment. Amended.
Point 3: Introduction can be condensed. Some details can go to discussion section (3rd para- lines 56-77)
Response 3: Thank you for your comment. Amended and transfer to discussion section in lines 347-352.
Point 4: Limitation: can be made more specific rather describing
Response 4: Thank you for your comment. We have added two paragraphs in limitation section (lines 385-387 and 391-397).

Reviewer 3 Report
I think that the authors faithfully revised this manuscript to reflect my comments. But the arrows in figure 1 are still weird and need to be modified further.
Author Response
Response to Reviewer 3 Comments Round 2
Thank you for your very positive and constructive feedback on our manuscript.
We have considered all your comments and suggestions (and also the comments made by the other reviewer) attempting to improve/refine the original manuscript.
Below you will find a point-by-point response to your comments (in red).
Point 1: I think that the authors faithfully revised this manuscript to reflect my comments.
Response 1: Thank you for your comment.
Point 2: But the arrows in figure 1 are still weird and need to be modified further.
Response 2: Thank you for your comment. The arrows were placed on the sides so as not to interfere with reading. Now we have centered the arrows and made them thinner.

Round 3
Reviewer 1 Report
This is as good as it's going to get.
Only one issue--please make sure that you update the exclusion criteria and other aspects based on what you have written here in clinicaltrials.gov.
Author Response
The changes have already been made in CT and are pending acceptance so that both protocols match. We would like to express our sincere appreciation for the invaluable comments you provided during the review process. Thank you very much for your valuable contribution.
Reviewer 2 Report
Nil
Author Response
We would like to extend my deepest gratitude for the invaluable comments you provided during the review process. Your contribution was truly valuable and we appreciate the time and effort you dedicated to this endeavor. Thank you very much.
Reviewer 3 Report
I support the publication of this manuscript.
Author Response
Thank you very much for your valuable comments in the review process.